# Insect endosymbiont proliferation is limited by lipid availability

Jeremy K Herren[1]*, Juan C Paredes[1], Fanny Schüpfer[1], Karim Arafah[2], Philippe Bulet[2,3], Bruno Lemaitre[1]*

[1]Global Health Institute, School of Life Sciences, Ecole Polytechnique Fédérale de Lausanne (EPFL), Lausanne, Switzerland; [2]Platform BioPark Archamps, Saint Julien en Genevois, France; [3]Université Joseph Fourier, AGIM FRE CNRS, La Tronche, France

**Abstract** *Spiroplasma poulsonii* is a maternally transmitted bacterial endosymbiont that is naturally associated with *Drosophila melanogaster*. *S. poulsonii* resides extracellularly in the hemolymph, where it must acquire metabolites to sustain proliferation. In this study, we find that *Spiroplasma* proliferation specifically depletes host hemolymph diacylglyceride, the major lipid class transported by the lipoprotein, Lpp. RNAi-mediated knockdown of Lpp expression, which reduces the amount of circulating lipids, inhibits *Spiroplasma* proliferation demonstrating that bacterial proliferation requires hemolymph-lipids. Altogether, our study shows that an insect endosymbiont acquires specific lipidic metabolites from the transport lipoproteins in the hemolymph of its host. In addition, we show that the proliferation of this endosymbiont is limited by the availability of hemolymph lipids. This feature could limit endosymbiont over-proliferation under conditions of host nutrient limitation as lipid availability is strongly influenced by the nutritional state.

## Introduction

Many insects harbor facultative bacterial endosymbionts, which despite not being required for host survival have important implications for host biology (*Wernegreen, 2012*). Two of the most prevalent and well-characterized facultative insect endosymbionts are *Wolbachia* and *Spiroplasma*, which are estimated to infect ~40% and 5–10% of all insects species, respectively (*Hackett and Clark, 1979*; *Duron et al., 2008*; *Hilgenboecker et al., 2008*). While *Wolbachia* principally resides intracellularly (*Dobson et al., 1999*; *Albertson et al., 2009*), *Spiroplasma* occupies an extracellular niche, proliferating mainly in the hemolymph that fills the body cavity of arthropods (*Sakaguchi and Poulson, 1961*; *Anbutsu and Fukatsu, 2006*). *Spiroplasma* and *Wolbachia* are both maternally transmitted and have developed unique strategies to colonize the germline of their female hosts for transmission to the next generation (*Frydman et al., 2006*; *Serbus and Sullivan, 2007*; *Herren et al., 2013*).

Facultative endosymbionts with strict maternal transmission, including *Wolbachia* and *Spiroplasma*, increase their prevalence in host populations by virtue of two strategies: (i) manipulating host reproduction to increase the fitness of infected hosts (*Werren and O'Neill, 1997*); (ii) inducing a direct increase in host fitness in a manner that is usually condition dependent, for example protecting hosts against different classes of parasites (*Hedges et al., 2008*; *Jaenike et al., 2010*; *Teixeira et al., 2008*). Protective endosymbionts of disease vectors may be useful for the control of vector borne disease, and they are increasingly being studied in this context (*Moreira et al., 2009*). While these interactions are clearly of importance, more fundamental features of facultative endosymbioses are poorly understood and frequently overlooked, including metabolic exchanges and the mitigation of host fitness costs.

Genome sequencing has indicated that endosymbiotic bacteria have highly reduced metabolic capacities and depend heavily on their hosts to provide them with a diversity of compounds needed for their sustained proliferation (*Klein et al., 2012*; *Moran et al., 2008*). However, the direct study of

*For correspondence:
jeremyherren@me.com (JKH);
bruno.lemaitre@epfl.ch (BL)

**Competing interests:** The authors declare that no competing interests exist.

**eLife digest** All animals host a large number of harmless microbes. Often the two partners involved in these interactions will depend on each other to thrive: microbes support important host functions and in return the host provides a safe place to live and a continuous supply of food. Many microbes that are intimately associated with animals have lost the ability to gain nutrients from sources other than their host and are unable to survive on their own. However, in many cases, the source and the type of nutrients provided to the microbes are unknown.

One of the most common microbial species found in insects is *Spiroplasma*. This microbe lives in very large numbers in the fluid that fills the body cavities of insects, called the hemolymph. The microbes are transmitted from mother to offspring, and in some circumstances can provide benefits to the insects; for instance, *Spiroplasma*-infested flies appear to be protected against infection by some parasites. Unfortunately, as it is difficult to study insect–microbe relationships, little else is known about the physiological interactions between these two species.

Herren et al. studied the association between *Spiroplasma* and the fly *Drosophila melanogaster*. Under normal conditions, *Spiroplasma* only reduces the life span of the infested fly. This indicates that *Spiroplasma* has a low impact on the general fitness of its host, only negatively affecting the survival and egg laying ability of old flies. When flies had limited access to nutrients, the number of *Spiroplasma* they carried was reduced, without the flies losing fitness. This suggests that *Spiroplasma* growth is dependent on something in the flies' diet.

To understand which nutrients are important for the growth of *Spiroplasma* in *Drosophila*, Herren et al. analyzed the hemolymph of flies and found that there are fewer fatty-molecules, called lipids, when nutrients are limited. Healthy flies carrying *Spiroplasma* also have fewer lipids in their hemolymph, suggesting that these are what *Spiroplasma* feed on. Indeed, inactivating a protein required by the fly to transport lipids to the hemolymph reduced the growth of *Spiroplasma* in these flies.

Herren et al. concluded that the growth of *Spiroplasma* inside its host is limited by the availability of lipids in the hemolymph. Since this is dependent on diet, the dependence on lipids couples the growth of *Spiroplasma* to the nutritional state of its host. Herren et al. speculate that this mechanism reduces the fitness cost of harboring the microbes and prevents the damaging consequence of an uncontrolled proliferation of the microbes. Moreover, *Spiroplasma*'s preference for lipids may explain why it helps to protect flies against parasitic infection, as many parasites also rely on lipids for their growth. Herren et al. suggest this strategy could also be used in other animal–microbe associations.

the metabolism of endosymbiotic bacteria is challenging, due to the high level of integration and interdependence between endosymbionts and their hosts. Therefore, despite a general, genome-centric understanding of the metabolic capacities of numerous endosymbionts, little is known about the nature of specific metabolites required for endosymbiont proliferation and the implications of metabolite acquisition by endosymbionts on host physiology and fitness.

Strict maternal transmission is expected to result in the evolution of endosymbionts that have minimized host fitness costs (*Werren and O'Neill, 1997*). Experimental studies are generally in line with this prediction, for example *Wolbachia* and *Spiroplasma* have relatively minor effects on host fitness (*Martins et al., 2010*; *Unckless and Jaenike, 2012*), however, fitness costs usually become apparent as hosts age (*Ebbert, 1991*; *Min and Benzer, 1997*; *Fry et al., 2004*). For endosymbionts that colonize the germline from the adult soma, it has been demonstrated that endosymbiont titers are positively correlated with transmission fidelity (*Dyer et al., 2005*; *Unckless et al., 2009*). In general, endosymbiont titers (and hence proliferation rates) will therefore be determined by a compromise between the need to minimize host fitness costs and maximize transmission fidelity (*Jaenike, 2009*). Since host fitness costs are most likely minimized by the limiting excessive endosymbiont proliferation, the factors that limit endosymbiont proliferation are of central importance for the biology of endosymbionts; however, few mechanisms that are capable of limiting endosymbiont proliferation have been identified (*Login et al., 2011*). The proliferation of bacteria is often controlled by host immune systems; however, it is notable certain endosymbionts, including *Spiroplasma*, are not susceptible to

host immune responses (*Herren and Lemaitre, 2011*) suggesting that other factors are likely to be of importance for limiting their proliferation.

In this study, we used the genetically tractable insect, *Drosophila melanogaster* and its endosymbiotic *Spiroplasma* (MSRO strain) to analyze the mechanisms that govern *Spiroplasma* proliferation and the effects of endosymbiont proliferation on host physiology. We find that under normal rearing conditions MSRO *Spiroplasma* (henceforth referred to as *Spiroplasma*) shortens the life span of its host, *D. melanogaster*. Interestingly, under nutrient limitation, where increased competition between *Spiroplasma* and its host could be expected, *Spiroplasma* proliferation is compromised with minimal effect on host fitness. We noted that under nutrient limitation, host hemolymph lipids decline significantly and that under normal rearing conditions, the first observable effect of *Spiroplasma*-infection on host physiology is a depletion of host lipids. We then used RNAi-based strategies to reduce the hemolymph lipid concentration and find that this inhibits *Spiroplasma* proliferation and extends the life span of *Spiroplasma*-harboring flies. We therefore demonstrate that: (i) specific hemolymph lipids are utilized by *Spiroplasma* and (ii) the availability of hemolymph-lipids limits the proliferation of *Spiroplasma*.

## Results

### The implications of harboring *Spiroplasma* on host fitness

We investigated the impact of harboring *Spiroplasma* on its host's fitness by measuring survival and egg production in both virgin and mated *Spiroplasma*-infected and uninfected female flies. When maintained on a rich *Drosophila* diet, flies-harboring *Spiroplasma* have a significantly shortened life span compared to flies that do not harbor *Spiroplasma* (*Figure 1A*, *Figure 1—figure supplement 1A*), which is in agreement with a previous study on *Drosophila willistoni* and WSRO *Spiroplasma* (*Ebbert, 1991*). Notably, the presence of *Spiroplasma* did not significantly affect the death rate until flies were about 21–25 days old but flies began to exhibit signs of *Spiroplasma*-induced pathology between 14 and 21 days, as demonstrated by decreased performance in climbing assays (*Figure 1B*). Prior to death, aged *Spiroplasma*-harboring flies exhibit an apparent lack of coordination and tremors. The increase in *Spiroplasma*-induced lethality and pathology in old flies correlates with the increase of *Spiroplasma* titers, observed by qPCR of whole flies (*Figure 1C*) and fluorescence microscopy of hemolymph (*Figure 1D*). Consistent with previous studies on other *Drosophila*-endosymbiotic *Spiroplasma* strains (*Anbutsu and Fukatsu, 2003*; *Haselkorn et al., 2013*), we noticed that *Spiroplasma* titers reached a plateau in old flies (age >28 days) suggesting that a factor might limit *Spiroplasma* proliferation at this stage. In light of the finding that *Spiroplama* reduces fly life span and the apparent trade-off between life span and reproductive output in *Drosophila* (*Partridge et al., 1987*; *Sgrò and Partridge, 1999*), we also compared the rate of egg production between *Spiroplasma*-infected and uninfected females. There was a twofold increase in the number of eggs laid by *Spiroplasma*-infected virgin flies compared to *Spiroplasma*-uninfected virgin flies over 14 days (*Figure 1E*). The number of eggs laid by *Spiroplasma*-infected mated flies over a 14-day period was similar to *Spiroplasma*-uninfected mated flies, however, *Spiroplasma*-infected mated flies laid an increased number of eggs in the first 2 days post-eclosion (*Figure 1—figure supplement 1B*). These results indicate that the presence of *Spiroplasma* stimulates egg production in virgin flies and also in mated flies over the first 2 days post eclosion, while causing a minor decline in egg production of mated flies at later time points. Collectively, these experiments demonstrate that (i) *Spiroplasma* has a low impact on the general fitness of its host, negatively affecting survival and egg laying only in old flies and (ii) that this decrease in host fitness correlates with higher *Spiroplasma* titers.

### Effects of nutrient limitation on host survival and *Spiroplasma* proliferation

We had initially speculated that the competition for resources between *Spiroplasma* and its host would be more conspicuous upon nutrient scarcity and that under these conditions *Spiroplasma* might have a more detrimental effect on host fitness. To test this hypothesis, we maintained adult flies on a nutrient poor diet. Under these nutrient-limiting conditions, *Drosophila* survival is significantly compromised, however it is important to note that this diet still contains sufficient nutritional content to support an entire *Drosophila* life-cycle (*Vijendravarma et al., 2012*). It is also worth mentioning that we only examined the effects of nutrient deprivation in adult flies and that for all the experiments conducted *Drosophila* larvae were raised under normal conditions. Surprisingly, *Spiroplasma*-infected and uninfected virgin flies had a similar life span (*Figure 2A*) and produced similar numbers of eggs under

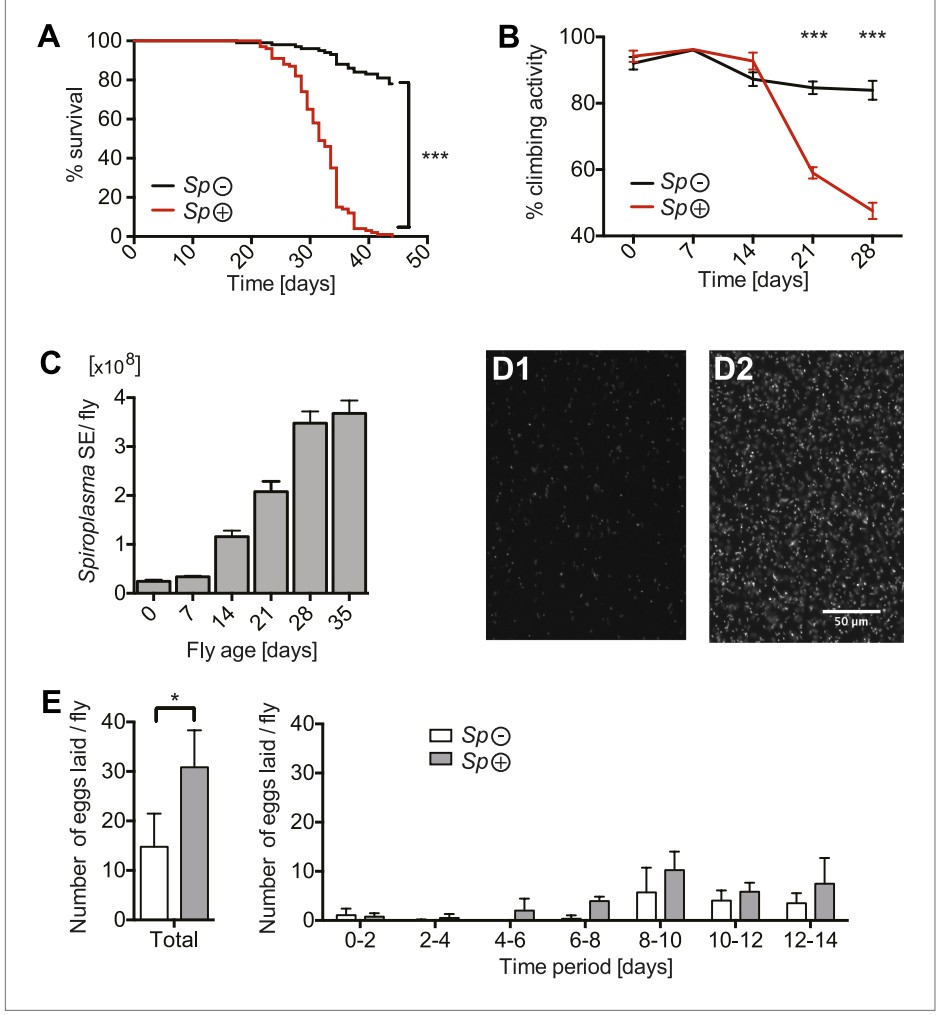

**Figure 1**. *Spiroplasma* proliferation is associated with life span reduction. (**A**) Life span of virgin flies-harboring *Spiroplasma* (*Sp* (+)) relative to control flies that do not harbor *Spiroplasma* (*Sp* (−)) when kept on a rich *Drosophila* diet. ***p<0.0001, N = 50 flies per condition. Shown is one representative experiment out of three independent experiments. (**B**) The climbing activity of virgin flies-harboring *Spiroplasma* (*Sp* (+)) relative to uninfected flies (*Sp* (−)) over time. ***p<0.0001, N = 20 flies per condition. Shown is one representative experiment out of three independent experiments. (**C**) qPCR quantification of the titers of *Spiroplasma* in virgin flies over aging. Values for each timepoint have at least three samples (five flies pooled per sample). Shown is one representative experiment out of three independent experiments. (**D**) Fluorescent microscopy images depicting SYTO-9 stained *Spiroplasma* in *Drosophila* hemolymph at 7 days (**D1**) and 21 days (**D2**) of fly age. (**E**) The number of eggs laid by virgin flies-harboring *Spiroplasma* (*Sp* (+)) relative to control flies that do not harbor Spiroplasma (*Sp* (−)), in total over 14 days (left panel) and in 2-day period over 14 days (right panel). In total, *Spiroplasma*-infected virgin flies laid significantly more eggs. *p=0.02. Shown is the mean ± SEM of data pooled from four independent experiments, N = 20 flies per experiment.

The following figure supplement is available for figure 1:

**Figure supplement 1**. The impact of *Spiroplasma* infection on survival and egg production by mated females on rich media.

nutrient-limiting conditions (*Figure 2B*). For mated flies under nutrient limitation, harboring *Spiroplasma* resulted in a minor reduction in survival relative to flies that did not harbor *Spiroplasma* (*Figure 2— figure supplement 1A*). *Spiroplasma*-infected mated flies produced more eggs than *Spiroplasma*-uninfected mated flies under nutrient-limiting conditions (*Figure 2—figure supplement 1B*). These findings suggest that, in contrast to our initial hypothesis, harboring *Spiroplasma* has a rather limited

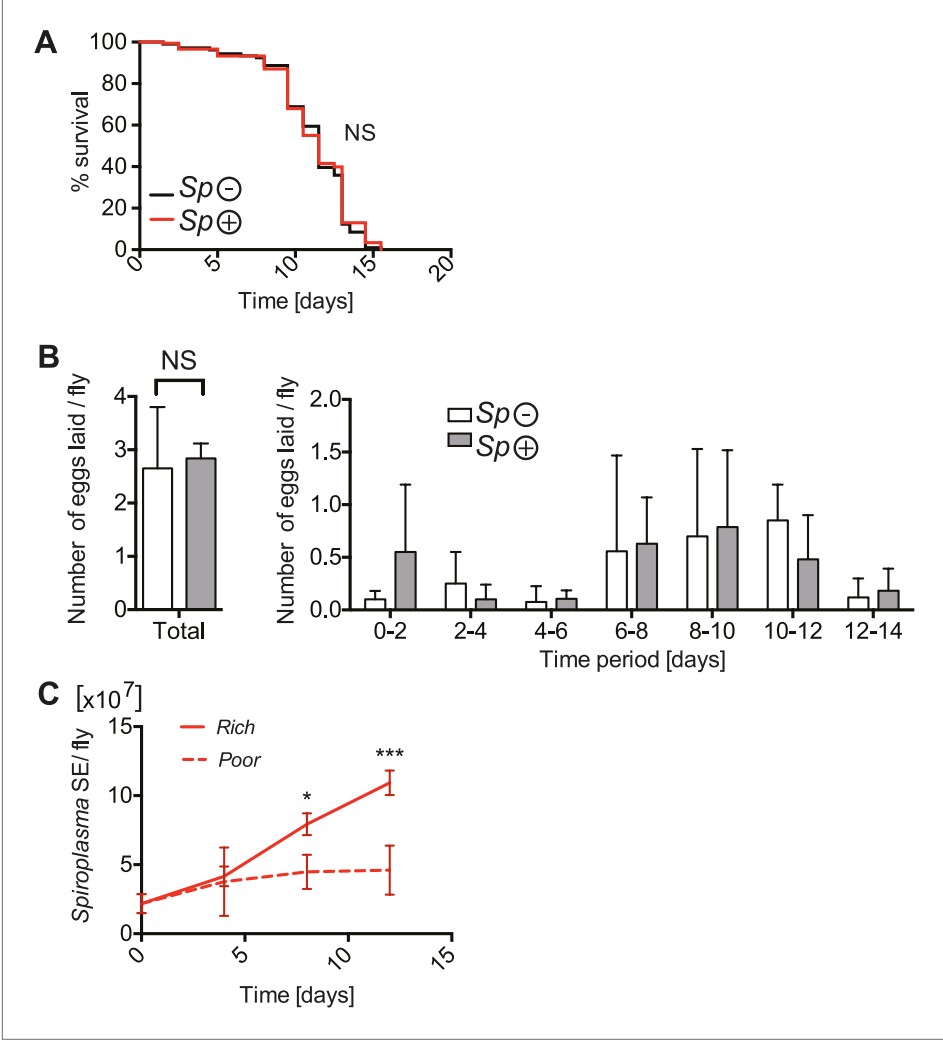

**Figure 2**. The implications of harboring *Spiroplasma* under host nutritional depravation. (**A**) Survival of virgin flies on a nutritionally poor diet. Flies-harboring *Spiroplasma* (*Sp* (+)) do not have significantly different mortality from flies that do not harbor *Spiroplasma* (*Sp* (−)). NS (p=0.9378). N = 50 flies per condition. Shown is one representative experiment out of three independent experiments. (**B**) The number of eggs laid by virgin flies-harboring *Spiroplasma* (*Sp* (+)) relative to control flies that do not harbor *Spiroplasma* (*Sp* (−)), in total over 14 days (left panel) and in 2-day period over 14 days (right panel). Overall, there is no significant difference in the number of eggs laid between *Spiroplasma*-infected and uninfected virgin flies under nutrient deprivation. NS (p=0.77). Shown is the mean ± SEM of data pooled from four independent experiments, N = 20 flies per experiment. (**C**) Quantification of *Spiroplasma* titers by qPCR reveals that virgin female flies maintained on the same nutritionally poor diet as in panel **A** have significantly lower *Spiroplasma* titers after 8 and 12 days than flies maintained on a rich diet. *p=0.015 and ***p=0.0002, respectively. Values for each time-point have at least three samples (five flies pooled per sample). Shown is the mean ± SEM of one representative experiment out of the three independent experiments that were conducted.

The following figure supplements are available for figure 2:

**Figure supplement 1**. The effects of nutrient deprivation on survival and egg production of *Spiroplasma*-infected mated females.

**Figure supplement 2**. *Spiroplasma* titers in fly hemolymph under nutrient deprivation.

fitness cost under nutrient-limiting conditions. Importantly, we observed that *Spiroplasma* proliferation in whole flies (*Figure 2B*) and hemolymph (*Figure 2—figure supplement 2*) was also significantly inhibited when flies are maintained on a nutrient poor diet. Thus, the inhibition of *Spiroplasma* proliferation in flies that are maintained under nutrient-limiting conditions could explain why *Spiroplasma* has limited fitness costs under these conditions. For consistency, and to facilitate the maintenance of flies under identical conditions, subsequent experiments were carried out on virgin females (unless otherwise specified).

## *Spiroplasma* proliferation is influenced by the nutrient composition of the diet

The results above suggest that *Spiroplasma* proliferation might be dependent on the availability of host factors that are nutritional state-dependent. To identify these factors, we examined the effect that maintaining uninfected flies for 12 days on a nutrient poor diet had on the concentration of metabolites in the hemolymph (where *Spiroplasma* reside). We found that raising flies on a nutrient poor diet resulted in significantly lower concentrations of protein, sterol and diacylglyceride (DAG, the main transport lipid in *Drosophila*) (*Figure 3A–C*), while levels of glucose and trehalose were not changed significantly (*Figure 3—figure supplement 1A,B*) and L-amino acids increased (*Figure 3—figure supplement 1C*). Thus, nutrient limitation led to a specific decline in hemolymphatic protein and lipid concentrations. We then complemented the nutrient poor diet with either inactivated yeast (rich in protein and lipids) or sucrose, we found that only inactivated yeast extract was able to recover the *Spiroplasma* proliferation rates observed on rich media (*Figure 4*). This indicates that *Spiroplasma* proliferation is not only affected by the caloric content of the food but by the composition of the diet. Taken together, these results lend support to the hypothesis that *Spiroplasma* proliferation is heavily dependent on hemolymph metabolite composition and that certain metabolites (e.g., lipids and protein) could play a more important role than others (e.g., sugars).

## *Spiroplasma* proliferation depletes lipids from the hemolymph and fat body

To gain a better insight into the relationship between *Spiroplasma* and host physiology, we monitored the effects of harboring *Spiroplasma* on the concentration of metabolites in the hemolymph of flies raised on a rich diet. We observed that the concentration of measured sugars was not changed and that the L-amino acid concentration was even increased in the presence of *Spiroplasma* (*Figure 5A–C*). We also found that the hemolymph of flies-harboring *Spiroplasma* had an increased concentration

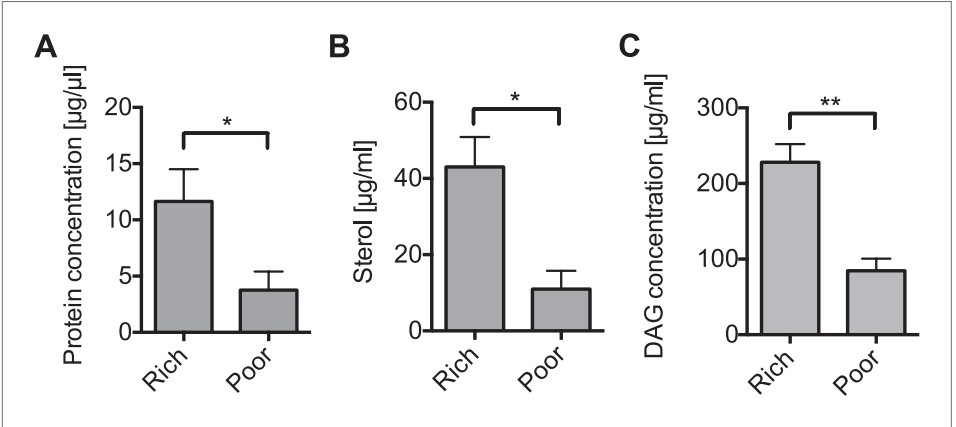

**Figure 3**. Nutrient deprivation depletes host lipids. (**A–C**) The protein (**A**), sterol (**B**), and DAG (**C**) concentration of the hemolymph of flies maintained on nutritionally poor diets for 12 days is significantly lowered relative to flies maintained on a nutritionally rich diet. Mean ± SEM of three independent experiments is shown, *p=0.038, *p=0.023, and **p=0.0018, respectively.
The following figure supplement is available for figure 3:

**Figure supplement 1**. The impact of nutrient deprivation on hemolymph metabolite concentrations.

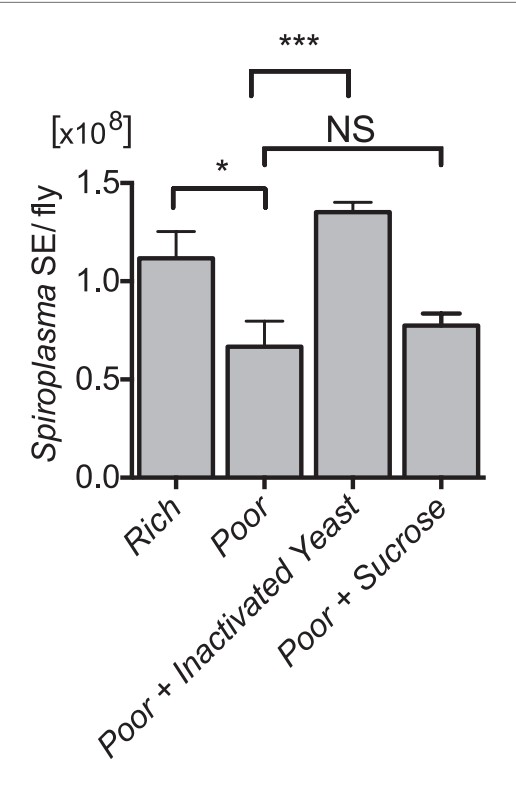

**Figure 4**. *Spiroplasma* proliferation is influenced by the nutrient composition of the diet. Quantification of *Spiroplasma* titers by qPCR reveals that complementing nutrient poor media with inactivated yeast results in a significant increase in *Spiroplasma* titers after 12 days. \*\*\*p=0.0003. In contrast, complementing nutrient poor media with sucrose does not significantly increase *Spiroplasma* titers. NS (p=0.5). Values are the mean ± SEM of at least four samples (five flies per sample). Shown is one representative experiment out of the three independent experiments that were conducted.

of proteins and sterol (*Figure 5D,E*). However, after centrifugation to remove bacterial cells, the protein and sterol concentration was no longer significantly different between *Spiroplasma*-infected and *Spiroplasma*-uninfected hemolymph samples. This suggests that *Spiroplasma* cells contain a substantial amount of proteins and sterol, but that the presence of this endosymbiont does not deplete either of these factors in the hemolymph. Notably, we observed that *Spiroplasma*-infected flies experienced a significant drop in the concentration of DAG within the hemolymph (*Figure 5F*). Importantly, this difference is not caused by the previously noted differences in the rate of egg production between *Spiroplasma*-infected and *Spiroplasma*-uninfected virgin flies, as a decrease in DAG was also observed in mated flies (*Figure 5—figure supplement 1A*), where *Spiroplasma* did not affect the number of eggs laid over 14 days (*Figure 1—figure supplement 1B*). We also observed that there was a more marked decline in DAG levels in 28-day-old *Spiroplasma*-infected relative to uninfected flies (*Figure 5—figure supplement 1B*), where *Spiroplasma* titers have nearly reached their maximum levels (*Figure 1C*) and egg production declines (*David et al., 1975*; *Partridge et al., 1987*). We then investigated the impact of *Spiroplasma* on metabolite storage in the fat body by quantifying the amount of triacylglyceride (TAG, the main storage lipid in *Drosophila*) and glycogen (the main storage carbohydrate in *Drosophila*). Quantifications of whole female flies (reflecting mainly insect fat body energy storage but also the energy contents of the ovaries) showed a decrease in the amount of TAG in *Spiroplasma*-infected flies compared to *Spiroplasma*-uninfected flies (*Figure 5G*), while the amount of glycogen was not affected by the presence of *Spiroplasma*

(*Figure 5—figure supplement 2*). Consistent with the decrease in TAG reserves, 12-day-old *Spiroplasma*-infected flies succumb more rapidly to acute starvation, in which flies are only given a source of water but no source of nutrition (*Figure 5H*). Since TAG stored in the fat body largely derived from hemolymph DAG (*Canavoso et al., 2001*), the decrease of TAG is likely to be the outcome of the depletion of hemolymphatic DAG by *Spiroplasma*. Thus, our analyses show that the proliferation of *Spiroplasma* in flies is associated with a specific depletion of hemolymph DAG as well as a decrease in the amount of fat body lipid storage. These findings, together with the observation that DAG is depleted under nutrient deprivation, suggest that DAG availability limits *Spiroplasma* proliferation.

### *Spiroplasma* proliferation is associated with the production of cardiolipin

Since the hemolymph derived from *Spiroplasma*-infected flies contained *Spiroplasma* and had reduced DAG, it is likely that *Spiroplasma* is not directly incorporating DAG into their membrane but rather metabolize DAG into another compound. The main classes of lipids that are present in *Spiroplasma* are phosphatidylglycerols, sterols, sphingolipids, and cardiolipins (*Freeman et al., 1976*). Cardiolipins are a class of lipids found exclusively in eubacteria and in the mitochondria of eukaryotic cells (*Hawthorne and Ansell, 1982*). In contrast to other major lipidic components of *Spiroplasma* membranes, cardiolipins are not present at detectable levels in the *Drosophila* hemolymph (*Carvalho et al., 2012*).

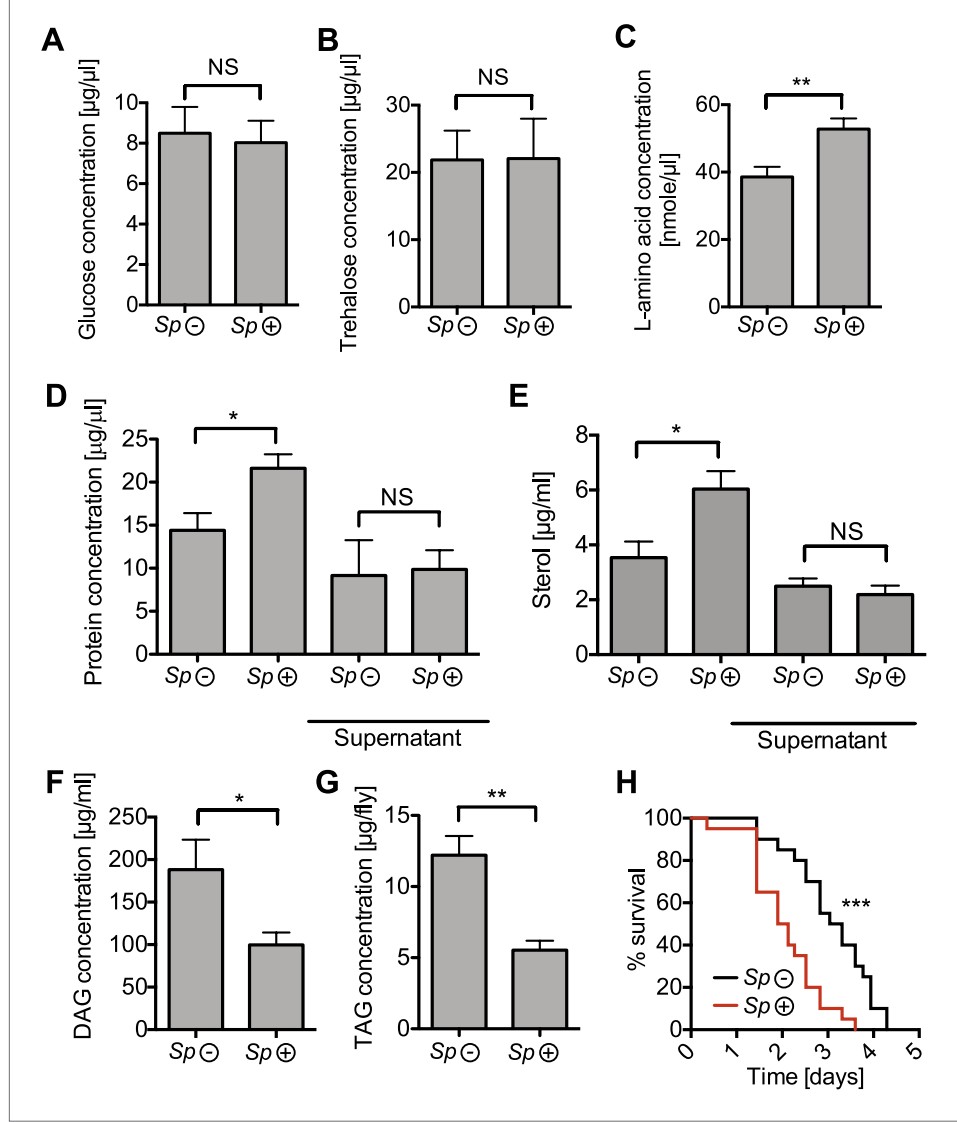

**Figure 5**. *Spiroplasma* infection depletes lipids of *Drosophila* maintained under normal conditions. Quantification of metabolites in flies that have been maintained on rich media for 12 days. Glucose (**A**), trehalose (**B**), and L-amino acid (**C**) concentration within the hemolymph of uninfected flies (*Sp* (−)) and *Spiroplasma*-infected flies (*Sp* (+)). L-amino acid concentration in the hemolymph is significantly higher in *Spiroplasma*-infected flies while glucose and trehalose concentrations remain unchanged. Mean ± SEM of three independent experiments is shown, NS (p=0.798 and p=0.977) and \*\*p=0.0056. (**D–E**) Quantifications of protein (**D**) and sterol (**E**) concentration in hemolymph from flies that harbor *Spiroplasma* (*Sp* (+)) and uninfected flies (*Sp* (−)). Hemolymph samples denoted as 'supernatant' have been subjected to an additional centrifugation to remove *Spiroplasma* cells, whereas all other hemolymph samples contain both *Spiroplasma* cells and hemolymph. Flies-harboring *Spiroplasma* have significantly higher total levels of protein and sterol in the hemolymph. Mean ± SEM of three independent experiments is shown, \*p=0.04 and \*p=0.037. After centrifugation to remove bacteria from the hemolymph, there was no longer any significant difference in protein and sterol concentrations between *Spiroplasma*-infected and uninfected hemolymph. Mean ± SEM of three independent experiments is shown, NS (p=0.881 and p=0.491, respectively). (**F**) Quantification of DAG content of hemolymph extracts from flies that harbor *Spiroplasma* (*Sp* (+)) and flies that do not (*Sp* (−)). \*p=0.0266. Mean ± SEM of three independent experiments is shown. (**G**) Quantification of whole-fly (reflecting mainly fat body) TAG levels in flies that harbor *Spiroplasma* (*Sp* (+)) and uninfected flies (*Sp* (−)). \*\*p=0.0043. Mean ± SEM of three independent experiments is shown. (**H**) Survival of flies subjected to an acute starvation after being maintained on rich media for 12 days. Flies-harboring *Spiroplasma* (*Sp* (+)) have
*Figure 5. Continued on next page*

*Figure 5. Continued*

significantly greater mortality rate than flies that do not harbor *Spiroplasma* (*Sp* (−)). ***p<0.0001. N = 20 flies per condition, shown is one representative experiment out of three independent.

The following figure supplements are available for figure 5:

**Figure supplement 1**. The impact of *Spiroplasma* proliferation hemolymph DAG concentration in mated flies and old flies.

**Figure supplement 2**. The impact of *Spiroplasma* proliferation on fat body glycogen stores.

Cardiolipins are comprised of four acyl chains, two phosphate groups and three glycerols, and therefore are made up of similar components to DAGs, which consist of two acyl chains and a glycerol. Analysis of the *Spiroplasma* genome reveals that *Spiroplasma* is nearly devoid of lipid metabolic capacities but possess genes involved in the synthesis of cardiolipin from the precursor DAG through a pathway involving DAG-3-phosphate and cytidine diphosphate-DAG (unpublished data). To investigate the production of cardiolipin by *Spiroplasma*, we conducted a MALDI time-of-flight mass spectrometry (MALDI-Tof-MS) analysis of lipid species in fly hemolymph and found that there were only peaks at *m/z* values that correspond to cardiolipin in the hemolymph of flies-harboring *Spiroplasma* (***Figure 6A***). A major ion at *m/z* 1425.05 was isolated and fragmented, which confirmed the presence of cardiolipin and revealed that the cardiolipins were comprised of C16:0 and C18:1 acyl chains (***Figure 6B***). In addition, we used liquid chromatography–tandem mass spectrometry (LC-MS/MS) to quantify the effect of *Spiroplasma* infection on the concentration of individual DAG species in *Drosophila* hemolymph. We observed that overall DAG concentration declined by 16.3%, however certain DAG species (e.g., C32:1 and C34:1 DAG) declined to a much greater extent (***Table 1***). It is notable that DAG species that decline to the greatest extent in the presence of *Spiroplasma* are those likely to contain (based on *Drosophila* fatty acid composition) one saturated (e.g., C14:0 or C16:0) and one mono-unsaturated (e.g., C16:1 or C18:1) medium-length acyl chain (***Shen et al., 2010***). The observation that *Spiroplasma*-generated cardiolipin contains the same configuration of one saturated and one mono-unsaturated medium-length acyl chain (predominately C16:0, C18:1) indicates that *Spiroplasma* most likely produce cardiolipin using fly hemolymph DAG. The transformation of DAG into cardiolipin offers an explanation for the observed decrease in hemolymph DAG levels in the presence of *Spiroplasma*.

## *Spiroplasma* acquires hemolymph-lipids prior to storage in the fat body

In *Drosophila*, dietary lipids are broken down in the gut lumen by lipases prior to absorption by intestinal cells (***Sieber and Thummel, 2012***). In the enterocytes, these compounds are used for the synthesis of DAG, which is packaged together with phosphoethanolamine, sterol, other minor lipids, and the apolipophorin protein (Lpp), to form lipoprotein particles. Lpp is produced in the fat body but travels to the gut where it gets loaded with lipids prior to trafficking throughout the body (***Palm et al., 2012***). Lpp is the main hemolymph lipid carrier, since more than 95% of the hemolymph lipids in *Drosophila* co-fractionate with Lpp (***Palm et al., 2012***).

Our results are consistent with *Spiroplasma* subverting and utilizing the lipids contained in hemolymph lipoprotein particles prior to their arrival at the fat body. This has the consequence of decreasing the observed levels of stored lipids. To rule out the possibility that *Spiroplasma* induces the mobilization of lipid stores, which could also decrease TAG levels, we quantified the effect of *Spiroplasma* on TAG levels in *AKHR* and *Bmm* double mutant flies. Adipokinetic hormone receptor (AKHR) and the Brummer lipase (Bmm) are components of two independent pathways that mobilize lipids from *Drosophila* fat bodies. Signaling through AKHR initiates mobilization of stored lipid in the insect fat body (***Gäde and Auerswald, 2003***), and Bmm is a TAG lipase involved in mobilization of stored lipid (***Grönke et al., 2005***). *AKHR[1];bmm[1]* double mutant flies exhibit an 'obese' phenotype because the fat bodies of these flies store lipids but are not subsequently able to mobilize or release lipids (***Grönke et al., 2007***). We found that fat body TAG levels in *AKHR[1];bmm[1]* double mutants that harbored *Spiroplasma* were still significantly lower after 12 days than flies with the same genotype that did not harbor *Spiroplasma* (***Figure 7***). While we cannot rule out other, as of yet uncharacterized, lipid mobilization pathways that could theoretically be activated by *Spiroplasma*, these results are still a strong

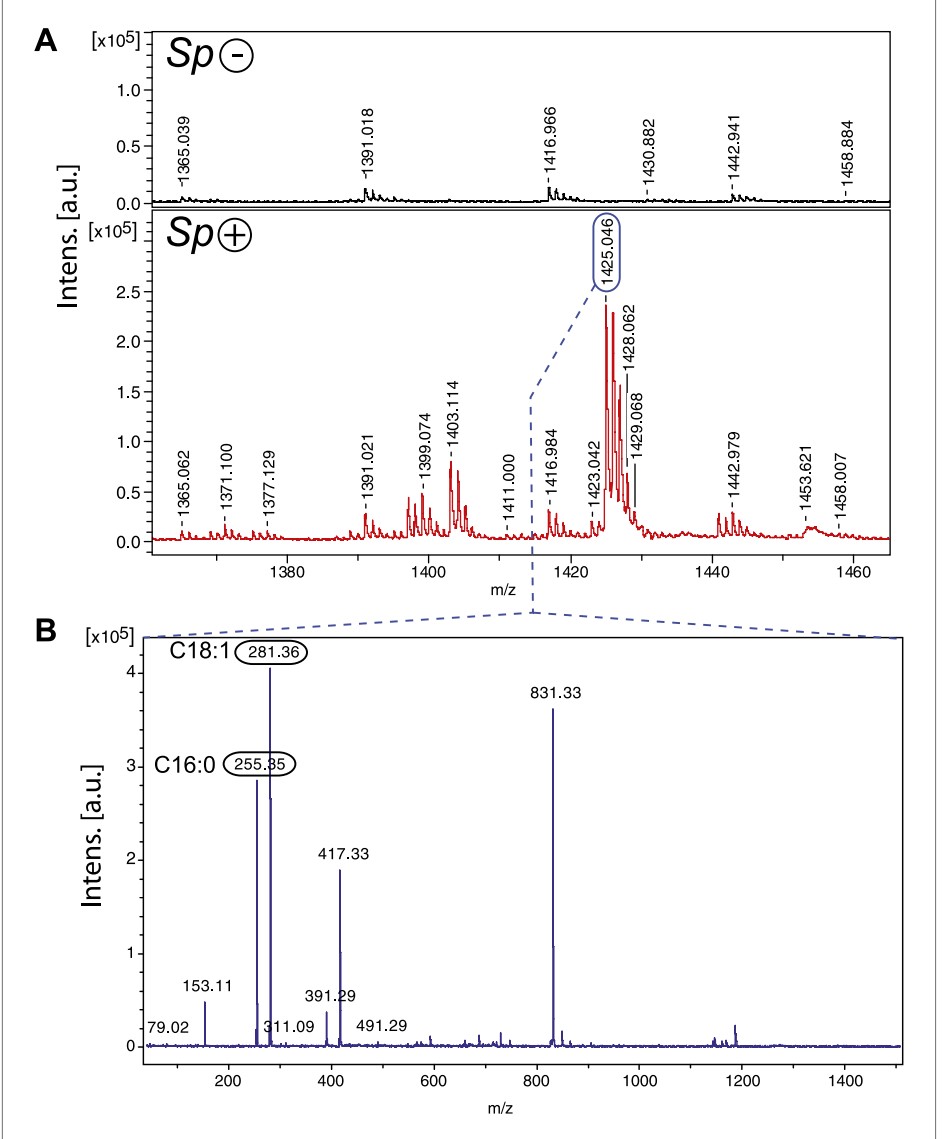

**Figure 6**. *Spiroplasma* produces cardiolipin in *Drosophila* hemolymph. (**A**) Negative MALDI-TOF/MS lipid profile of hemolymph from *Spiroplasma*-uninfected (top) and *Spiroplasma*-infected flies (bottom). The *m/z* signal peaks in the 1380–1460 range of *Spiroplasma*-uninfected hemolymph do not correspond to m/z values of cardiolipin, whereas the peaks in this region for *Spiroplasma*-infected hemolymph profile (e.g., 1403.11, 1425.05) do correspond to cardiolipin. (**B**) The isolation and fragmentation of the *m/z* 1425.05 parent ion resulted in the generation of daughter ions with peaks at *m/z* 281.36 and 255.35 that have been characterized as oleic acid (C18:1) and palmitic acid (C16:0) using Lipid MS Predict software with an error tolerance set to 0.1 u.m.a. The peak at *m/z* 79.02 detected in the same experiment corresponds to a phosphate ion. Six additional peaks (m/z 153.11, 311.09, 391.29, 417.33, 491.29 and 831.33) were also detected corresponding to phosphatidyl moieties and the cardiolipin 'backbone' (***Hsu and Turk, 2006***). Altogether, this indicates that the molecular ion corresponds to the cardiolipid species [M-2H+Na]⁻ *m/z* 1425.05.

indication that *Spiroplasma* lipid acquisition most likely occurs in the hemolymph before the point of lipid entry into the fat body stores.

## *Spiroplasma* proliferation depends on hemolymph lipid availability

We have observed that: (i) *Spiroplasma*'s proliferation is affected by host nutrient limitation and (ii) *Spiroplasma* consumes the hemolymph DAGs. These findings lead us to hypothesize that the proliferation of *Spiroplasma* could be constrained by the availability of lipids in the hemolymph, which

**Table 1.** Characterization and quantification of hemolymph of DAG species

| Component name | *Sp* − (µg/ml) | *Sp* + (µg/ml) | % Of total mass decline |
|---|---|---|---|
| C28:0 DAG | 13.4 | 12.3 | 7 |
| C28:1 DAG | 0.8 | 0.6 | 1.2 |
| C30:2 DAG | 0.7 | 0.5 | 1.2 |
| C30:0 DAG | 0.1 | 0.2 | −1.0 |
| C32:3 DAG | 1.2 | 1.1 | 0.7 |
| C32:2 DAG | 23.1 | 21.8 | 8.2 |
| C32:1 DAG | 19.8 | 15.1 | 30.1 |
| C34:2 DAG | 11.5 | 10.8 | 4.9 |
| C34:1 DAG | 9.7 | 6.9 | 19.3 |
| C34:0 DAG | 1.4 | 0.5 | 5.6 |
| C36:4 DAG | 0.8 | 0.5 | 1.8 |
| C36:3 DAG | 2.9 | 2.2 | 5.3 |
| C36:2 DAG | 4.2 | 2.9 | 9.3 |
| C36:1 DAG | 0.8 | 0.9 | −0.3 |
| C36:0 DAG | 1.7 | 0.7 | 6.9 |
| | 92.1 | 77 | |

Quantification of the absolute concentration of individual DAG species in the hemolymph of *Spiroplasma*-uninfected (*Sp* (−)) and *Spiroplasma*-infected (*Sp* (+)) mated flies by LC-MS/MS. The % of total mass decline reflects the percentage of the total decline between *Spiroplasma*-uninfected and infected samples (a total of 15.1 µg/ml or 16.3%) that can be attributed to each DAG species. It is notable that C32:1 and C34:1 DAG species decline to a greater extent than other common DAG species such as C28:0 and C32:2. This suggests that *Spiroplasma* is preferentially incorporating DAGs that have one saturated and one mono-unsaturated acyl chain. Notably, C34:1 DAGs are likely to be made up of oleic (C18:1) and palmitic (C16:0) acids, which have exactly the same acyl chains that were identified in *Spiroplasma*-generated cardiolipins (**Figure 6B**).

is known to be heavily dependent on the insect nutritional status (***Canavoso et al., 2001***). To further investigate the lipid-centric metabolic interplay between *Spiroplasma* and *Drosophila,* we monitored *Spiroplasma* proliferation in flies with a reduced capacity for inter-organ lipid transport and hence decreased hemolymph lipid concentrations. A recent study has shown that RNAi-mediated knockdown of *Lpp* in the fat body resulted in a decrease in circulating lipoprotein particles and a blockage of lipid export from the gut of *Drosophila* larvae (***Palm et al., 2012***). Based on these findings, we used a similar RNAi strategy to knockdown *Lpp* in the fat body of adult flies. In this experiment, we specifically expressed *Lpp-RNAi* in the fat bodies of adult flies using a flippase-mediated activation strategy (***Marois and Eaton, 2007***). We established that RNAi-mediated knockdown of *Lpp* does not significantly decrease the overall levels of protein in the adult hemolymph (***Figure 8—figure supplement 1***) but does decrease adult hemolymph-lipids, as shown by quantification of hemolymph DAG and sterol (***Figure 8A,B***). We then investigated the effect of this lipid reduction on the *Drosophila–Spiroplasma* interaction. We found that under conditions of *Lpp* knockdown, *Spiroplasma* proliferation was severely inhibited resulting in lower *Spiroplasma* titers in flies after 14, 21, and 28 days of aging (***Figure 8C***). To rule out any possibility that the genetic background of *Lpp-RNAi* flies was causing the observed decrease in *Spiroplasma* titers, we quantified *Spiroplasma* titers in the absence of activation of *Lpp-RNAi* by heat-shock and found no significant difference between flies containing the *Lpp-RNAi* construct and those that did not (***Figure 8—figure supplement 2A***). In addition, we used an independent *Lpp-RNAi* construct with another fat body specific driver and activation strategy to confirm that the decrease in *Spiroplasma* titers was specifically caused by RNAi-mediated *Lpp* knockdown (***Figure 8—figure supplement 2B***). It is noteworthy that *Spiroplasmas* are not known to have the capacity to utilize proteins as nutrient sources (***Chang and Chen, 1983***), supporting the claim that lipids carried by Lpp (and not the Lpp protein itself) are the factors required for the proliferation of *Spiroplasma*. Another striking effect of the RNAi-mediated knockdown of *Lpp* was strongly diminished *Spiroplasma*-induced old age mortality (***Figure 8D***).

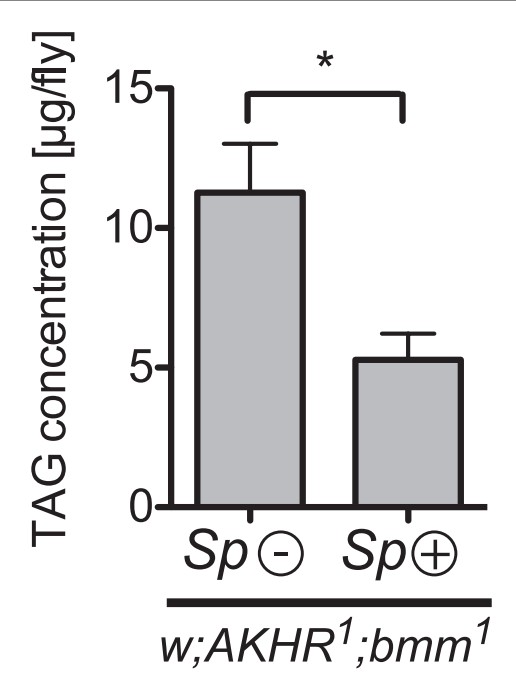

**Figure 7**. *Spiroplasma*-induced lipid depletion is not caused by the mobilization of stored lipids. TAG levels in *AKHR¹;Bmm¹* double mutants that harbor *Spiroplasma* (*AKHR¹;Bmm¹ Sp* (+)) relative to the same genotype without *Spiroplasma* (*AKHR¹;Bmm¹ Sp* (−)). *p<0.0153. Flies were maintained on a rich *Drosophila* diet for 12 days prior to TAG analysis. Mean ± SEM of three independent experiments is shown.

## Discussion

We have demonstrated that *Spiroplasma* subverts specific host lipids and that its proliferation is limited by the availability of host hemolymph-lipids. This finding is based on several observations: (i) *Spiroplasma* proliferation rate is decreased in the hemolymph of nutrient deprived flies and old flies, two conditions in which hemolymph–lipid concentration is decreased; (ii) the proliferation of *Spiroplasma* depletes hemolymph and fat body lipids; and (iii) a genetically induced reduction in Lpp-lipids inhibits *Spiroplasma* proliferation.

Natural selection is expected to favor vertically transmitted endosymbionts with adaptations that minimize fitness costs to their hosts (***Werren and O'Neill, 1997***). We hypothesize that *Spiroplasma's* dependence on hemolymph-lipid availability for proliferation could be of adaptive significance, since it would enable *Spiroplasma* to limit its proliferation in the face of host nutrient deprivation, and therefore avoid the costly depletion of its host's energy and other vital metabolites. If the proliferation rate of *Spiroplasma* were to be determined by metabolites that do not decline under nutrient-limiting conditions, for example sugars and L-amino acids, then *Spiroplasma* proliferation would not be limited under host nutritional deprivation. Such a lack of restriction could result in greater negative effects on its host's fitness. For example, we might expect a striking decrease in reproductive output or survival, which was not observed for *Spiroplasma*-infected flies maintained on a nutrient poor diet. We only observed that mated flies-harboring *Spiroplasma* had increased mortality after about 7 days under nutrient-limiting conditions, and this is not likely to have a major impact on host fitness since increased mortality occurred after the production of eggs had stopped.

Under normal conditions, *Spiroplasma's* significantly decreases the life span of its host *Drosophila* (also shown in ***Ebbert, 1991***). However, aspects of this *Spiroplasma*-induced pathogenesis, including the late onset of symptoms, suggest that *Spiroplasma* may be employing strategies to minimize host fitness costs. *Spiroplasma's* capacity to utilize available host lipids for proliferation results in depleted hemolymph-lipids and reduced fat body stores. This depletion also has a fitness consequence for flies, since *Spiroplasma*-infected flies die more rapidly when subjected to a period of acute starvation, but it is possible that *Spiroplasma's* reliance on these lipids might be less detrimental than usage of other more critical metabolites that would impact host fitness more directly.

*Drosophila* fitness and longevity are linked to egg production rates (***Partridge et al., 1987***), which can be affected by the presence of facultative endosymbionts (***Fast et al., 2011***). On nutrient rich media, we found that despite increasing early egg production *Spiroplasma* infection status did not affect the total number of eggs laid over a 14-day period. The *Spiroplasma*-induced early increase in egg production has been described previously for other *Spiroplasma* strains (***Ebbert, 1991***; ***Martins et al., 2010***). While the mechanistic basis of this increase is unknown, it offers a possible explanation for the greater number of eggs laid by *Spiroplasma*-infected flies on nutrient poor media, where the majority of eggs laid by all flies are in the first 2 days post-eclosion. Under rich nutritive conditions, we observed that *Spiroplasma* increased by twofold the number of eggs laid by virgin flies over a 14-day period. This finding is suggestive of a *Spiroplasma*-mediated disruption in the balance between egg retention and laying of virgin flies. One could speculate that *Spiroplasma* interferes with the signaling pathways (e.g., Juvenile Hormone and/or Ecdysone), which have previously been shown to regulate the production of eggs (***Soller et al., 1999***).

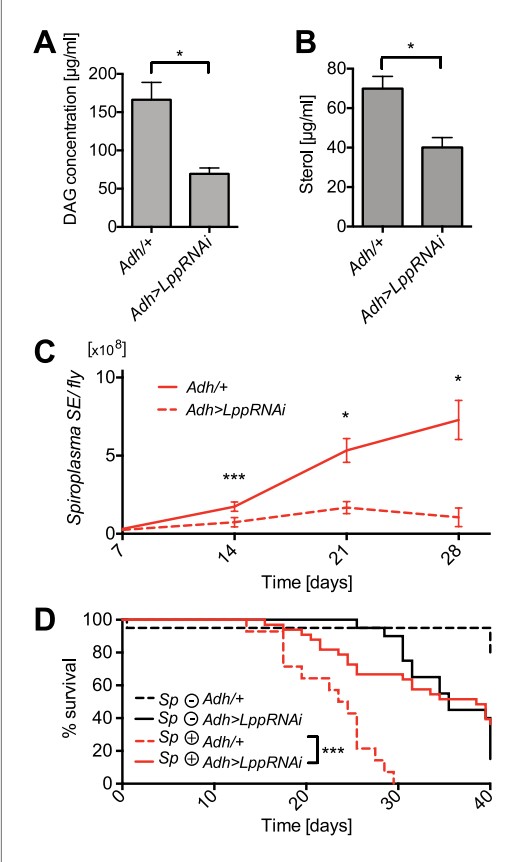

**Figure 8**. Lpp-lipids are required for *Spiroplasma* proliferation. (**A**) Quantification of DAG levels in hemolymph of 12-day-old flies 8 days after knockdown of *Lpp* by RNAi. Mean ± SEM of three independent experiments is shown, *p=0.015. (**B**) Quantification of sterol concentration in hemolymph of 12-day-old flies 8 days after knockdown of *Lpp* by RNAi. *p=0.0125. Mean ± SEM of three independent experiments is shown. (**C**) *Spiroplasma* titers quantified by qPCR in flies which have *Lpp* expression knocked down by RNAi relative to control flies. *Spiroplasma* titers were quantified at 7, 14, 21, and 28 days after activation of *RNAi*. Mean ± SEM of at least three samples is shown (five flies pooled per sample). ***p=0.0005 (14 days), *p<0.01 (21 days), and *p<0.01 (28 days). Shown is one representative experiment out of three independent experiments. (**D**) The survival of *Spiroplasma*-infected (*Sp* (+)) and *Spiroplasma*-uninfected (*Sp* (−)) flies with or without *RNAi*-mediated *Lpp* knocked down. ***p<0.0001, N = 18 flies per condition. Shown is one representative experiment out of three independent experiments.

The following figure supplements are available for figure 8:

**Figure supplement 1**. Hemolymph protein is not decreased by Lpp depletion.

**Figure supplement 2**. *Spiroplasma* titers are decreased by Lpp depletion.

We observed a decrease in *Spiroplasma* proliferation in flies after 28 days. A late decline in *Spiroplasma* proliferation has been demonstrated for a number of other *Spiroplasma* strains (*Anbutsu and Fukatsu, 2003*; *Haselkorn et al., 2013*). It is notable that flies aged for 28 days also experience a decline in DAG concentration, which is likely to explain the declining *Spiroplasma* proliferation rate. We suspect that *Spiroplasma* proliferation over the life of flies results in elevated *Spiroplasma* titers that ultimately deplete host hemolymph lipid and constrain further proliferation. In the MSRO-*Spiroplasma* system, this decline in proliferation does not appear to limit *Spiroplasma*-induced mortality, however, other *Drosophila*–*Spiroplasma* strains appear to reach maximal titers at earlier time points and therefore this is an aspect of the endosymbiont's biology could play a more important role in limiting fitness costs in other systems.

We noted that *Spiroplasma*-infected flies had higher levels of cardiolipin, suggesting that *Spiroplasma* synthesize cardiolipin, most likely using DAG acquired from host lipoproteins. We observed that *Spiroplasma* specifically depleted host DAG species that contained one saturated and one mono-unsaturated acyl chain, which was also the most abundant acyl chain configuration in *Spiroplasma*-produced cardiolipin (which contained C16:0 and C18:1 acyl chains). The basis of this acyl chain specificity is not known, but it could be linked to biophysical properties of cardiolipins in the highly curved *Spiroplasma* membrane. The utilization of DAG as a precursor for cardiolipin synthesis is supported by the annotation of the *Spiroplasma* genome (unpublished data), which enabled the identification of a biosynthetic pathway similar to that described in *Mycoplasma synoviae* and *Mycoplasma hyopneumoniae* for the synthesis of cardiolipin from acylglycerols through DAG-3-phosphate and cytidine diphosphate-DAG (*Arraes et al., 2007*). A number of studies have demonstrated links between cardiolipin and cell death (*Gonzalvez and Gottlieb, 2007*), which raise the possibility that cardiolipin produced by *Spiroplasma* could be involved in its pathological effects in old flies.

The other lipids that comprise a smaller fraction of Lpp's cargo, including sterols and sphingolipids (*Palm et al., 2012*) might also be important for *Spiroplasma* proliferation but these lipids would most likely be incorporated unchanged into the membrane of *Spiroplasma.* Of the other lipid classes contained in lipoproteins, sterols are of particular interest as they have been shown to be required for proliferation of all species of

*Spiroplasma*, and to be highly abundant in the *Spiroplasma* plasma membrane (*Freeman et al., 1976*). Amongst bacteria, this requirement for sterol is unique to mollicutes (*Dahl, 1993*). Since insects are auxotrophic for sterols (*Canavoso et al., 2001*), their concentrations are determined by dietary uptake. Although it is therefore possible that the sterol availability could play a key role in coordinating *Spiroplasma* proliferation with host nutritional state, we were unable to recover normal *Spiroplasma* proliferation rates when flies were maintained on nutrient poor media complemented with sterol (unpublished data). This finding suggests that, while sterol might be important for *Spiroplasma* proliferation, it is not sufficient to induce proliferation under conditions of nutrient limitation.

The strategies employed in different insect endosymbioses to limit over-proliferation of endosymbionts are not well characterized. We have proposed a model in which *Spiroplasma's* dependence on host hemolymph-lipid availability could limit its over-proliferation, primarily in the face of host nutritional scarcity. This mechanism could be important for the controlling proliferation of diverse extracellular endosymbionts. Different mechanisms are likely to be involved in intracellular endosymbiont control. For example, proliferation and localization of the obligate *Sitophilus* primary endosymbiont (SPE) is controlled by the expression of a specific antimicrobial peptide from its host, *Sitophilus* weevils (*Login et al., 2011*). In aphids, studies suggest the activation of the host lysosomal system is involved in controlling titers of the obligate intracellular endosymbiont, *Buchnera* (*Nishikori et al., 2009*).

The metabolic exchanges between endosymbiotic bacteria and their arthropod hosts are generally not well understood, partly due to challenges associated with high levels of integration and interdependence between partners. Most of the available information is indirect, coming from the study of endosymbiont genomes, which reveals limited metabolic capacities and high levels of dependence on hosts (*Zientz et al., 2004*; *Gosalbes et al., 2010*). There are a number of studies that examined metabolic exchanges between hosts and obligate endosymbionts, which provide their hosts with one or more vital metabolite that is missing from the host's diet (*Douglas, 1998*; *Russell et al., 2013*). These studies usually consider metabolic transfers from endosymbiont to host and not in the reverse direction (*Dale and Moran, 2006*). We have identified a transfer of metabolites from host to endosymbiont, showing that *Drosophila* Lpp-lipids are used by *Spiroplasma* for proliferation and more specifically provide evidence indicating that host-derived DAG is converted to cardiolipin by *Spiroplasma*.

A number of recent studies have revealed that facultative endosymbionts protect their hosts from parasites and pathogens (*Oliver et al., 2003*; *Hedges et al., 2008*; *Teixeira et al., 2008*). Endosymbiotic *Spiroplasma* has been implicated in a number of cases, including protecting diverse hosts from various eukaryotic parasites including parasitoid wasps, parasitic nematodes, and fungi (*Jaenike et al., 2010*; *Xie et al., 2010*; *Łukasik et al., 2012*). We speculate that *Spiroplasma*-mediated protection could be linked to lipid utilization. Indeed, many parasitoid wasps are unable to synthesize fatty acids and their development requires the acquisition of host lipids (*Visser et al., 2010*). Thus, sequestration of lipid by *Spiroplasma* might limit availability to any parasites or pathogens that occupy the same niche.

Here, we have shown that the rate of *Spiroplasma* proliferation and onset of fly mortality are decreased upon depletion of hemolymph lipids. It is noteworthy that a variety of other microorganisms known to proliferate in *Drosophila* hemolymph and to cause pathogenesis such as *Erwinia carotovora strain 15*, *Listeria monocytogenes*, *Candida albicans*, and *Enterococcus fecalis* do not appear to have attenuated virulence when hemolymph lipids are depleted (unpublished data). Many of these pathogens are free-living and are likely to have retained well-developed metabolic capacities (including the capacity to synthesize lipids). It is interesting to speculate that increased reliance on host provision of lipids is part of a suite of adaptations that facilitate the evolution of chronic, low-virulence infection strategies. It is notable that a number of pathogens that have the capacity to form chronic infections in humans including *Mycobacterium tuberculosis* and *Chlamydia trachomatis* have been shown to be heavily dependent on lipids acquired from their hosts (*Ehrt and Schnappinger, 2007*; *Robertson et al., 2009*). Thus, the findings discussed here for endosymbionts could be of more general importance for host–microbe interactions.

## Materials and methods

### Fly stocks and handling

We used a wild-type *Oregon-R* (*OR^R*) fly stock that harbors MSRO *Spiroplasma* (*Pool et al., 2006*; *Herren and Lemaitre, 2011*) but not *Wolbachia*. The *w;AKHR^1;bmm^1* stocks used have been described (*Grönke et al., 2007*). RNAi-mediated knockdown of *Lpp* was achieved using a pFRiPE-mediated

inducible RNAi element (*Marois and Eaton, 2007*), placed in the presence of a heat-shock inducible flippase and an *Adh-GAL-4* driver that is mostly active in the fat body. The strategy and stocks used are analogous to a previously published study, except that we induced RNAi in adults, as opposed to larvae (*Palm et al., 2012*). We induced the flippase by heat-shock (1.5 hr at 37°C in a water bath) in 4- to 5-day-old adult flies. This results in the excision of an upstream spacer region and activation of the UAS-*Lpp-RNAi* construct driven by *Adh-GAL-4* and the silencing of *Lpp* in the fat body, specifically at the adult stage. An additional strategy used to knockdown the expression of *Lpp* involved the *C564-GAL-4* driver (also mostly active in the fat body) in conjunction with *tubulin-gal80ts*, a temperature-sensitive repressor of *GAL-4* expression, which blocks GAL-4 expression at 18°C but not 29°C (*McGuire et al., 2003*). Flies that contained both the *C564-GAL-4, tubulin-gal80ts*, and UAS-*Lpp-RNAi(46)* elements were maintained at 18°C and then shifted to 29°C as adults to induce UAS-*Lpp-RNAi(46)* and knockdown *Lpp* expression. The UAS-*Lpp-RNAi(46)* stock is TRiP #HM05157 originating from the transgenic RNAi project at the Harvard Medical School. Since *Spiroplasma MSRO* is vertically transmitted and kills male embryos, *Spiroplasma*-infected stocks were generated in several steps: (1) Crossing an infected *OR^R* female with males carrying appropriate balancer chromosomes (2) Crossing the balanced female progeny with males of the genotype of interest (either *w;AKHR;Bmm*, *w,HS-FLP;adh-GAL-4/bcg* or *Dipt-GFP,C564-GAL-4;tub-GAL80ts*) (3) Several back-crosses were then carried out, resulting in a homozygous *Spiroplasma*-infected female with the appropriate genotype. These stocks were then maintained by crossing females with non-infected males of the same genotype. *Spiroplasma*-infected females carrying a *GAL-4* construct were then crossed with males carrying *UAS-RNAi* constructs or controls (*w* background). All flies were maintained at 25°C unless otherwise specified. The density of animals per vial was equilibrated between *Spiroplasma*-infected and *Spiroplasma*-uninfected stocks for development under similar levels of larval competition. Unless otherwise specified, all flies used were females and virgins. For survival experiments, counts were made every 24 hr and flies were transferred to new tubes every 3–4 days (2 days for mated flies). Climbing assays were carried out as described (*Barone and Bohmann, 2013*). Acute starvation assays have been described previously (*Grönke et al., 2007*). To quantify the number of eggs laid, flies were collected immediately post-eclosion and maintained in individual *Drosophila* vials that each contained five flies. Eggs were counted every 48 hr over a 14-day period, and the number of eggs laid per 48 hr per fly was calculated (correcting for any fly mortality). For experiments that required mated flies, males (three for every five females) were placed in the *Drosophila* vials and then removed after 7 days, males that died prior to this were replaced.

### *Drosophila* media
Flies were raised and maintained on a standard cornmeal-agar diet, referred to as 'rich media'. Normal media contain 4.5 g agar, 58.8 g inactivated yeast (Springaline BA95/0; Biospringer, Milwaukee, WI, USA), 35 g maize flour (Farigel Maize; Westhove, Ennezat, France), 34.8 ml of 1:1 mix of grape and multi-fruit juice (approximately 8.2 g of sugar), 3.6 ml of propionic acid, and 18 ml of a 10% solution of methyl paraben in 85% ethanol per 600 ml of water. For nutrient deprivation, adults were maintained on a restrictive diet referred to as 'poor media', which contains 9 g agar, 1.9 g inactivated yeast, 7.5 g maize flour, 4.5 g sucrose, 9 g glucose, 0.3 g MgSO4, 0.3 g CaCl$_2$, 3.6 ml propionic acid, and 18 ml of a 10% solution of methyl paraben in 85% ethanol per 600 ml of water (*Vijendravarma et al., 2012*). Poor media complemented with inactivated yeast contain an additional 34 g of inactivated yeast. Poor media complemented with sucrose contain an additional 35.5 g of sucrose. In all cases, larvae were raised on rich media.

### Imaging
To observe *Spiroplasma* in fly hemolymph, flies were dissected on microscope slides in 5 μl PBS containing 0.02 mM SYTO9 (Invitrogen, Carlsbad, CA, USA). Slides were then mounted and observed on an Axioimager Z1 (Zeiss, Oberkochen, Germany). Images were captured with an Axiocam MRn camera and Axiovision software.

### DNA extraction and qPCR
We extracted DNA from five flies per sample. The DNA extraction and quantitative PCR protocols have been previously described (*Anbutsu and Fukatsu, 2003*; *Herren and Lemaitre, 2011*). To determine the absolute number of bacteria per extraction, we extracted infected fly hemolymph and used fluorescence microscopy to calculate the concentration of *Spiroplasma* cells stained using SYTO9

(as described above). A dilution series of known concentrations of *Spiroplasma* cell equivalents (SE) was then combined with five uninfected flies prior to DNA extraction and qPCR, which enabled us to generate a calibration curve. In subsequent analyses, to account for differences between qPCR runs, we always used a positive control of known *Spiroplasma* concentration. The results for all experiments involving *Spiroplasma* titers are given in SE per fly, which represents the absolute quantities of *Spiroplasma* per fly. A host gene, RPS17, was also always quantified to verify the quality of the extraction but did not use this value in the analyses. To quantify bacteria in hemolymph samples, 0.5 μl of hemolymph (collected as described for metabolite analyses) was diluted in 300 μl cell lysis buffer prior to DNA extraction and qPCRs as previously described (*Herren and Lemaitre, 2011*).

## Metabolite analyses

*Drosophila* hemolymph was collected from flies individually using a Drummond nanoject and pulled capillary needle. Metabolic quantifications are given as the mass (μg) of metabolite per μl or ml of *Drosophila* hemolymph. For each hemolymph sample, we collected 2 μl (~25 flies), which was then diluted in 100 μl H$_2$O prior to subsequent analyses. For all analyses hemolymph samples were centrifuged at 13,000×*g* for 2 min to remove *Drosophila* cells. Protein concentration was determined using a Bradford assay (Bio-Rad, Hercules, CA). Glucose and trehalose concentration was quantified using glucose HK kit (Sigma-Aldrich, St. Louis, MO, USA). Samples were treated with or without trehalase (11 mU, Sigma-Aldrich) overnight at 37°C, and trehalose levels were obtained by subtracting the amount of free glucose in the untreated sample from the total glucose present in the sample treated with trehalase. L-amino acid concentration was quantified by a coupled enzyme reaction, using the L-amino acid quantitation kit (Sigma-Aldrich). Hemolymph DAG and fat body TAG were analyzed using a coupled colorimetric assay (*Hildebrandt et al., 2011*). Sterols were quantified using the Amplex Red Cholesterol Assay kit (ThermoFisher Scientific, Waltham, MA, USA), which detects primarily cholesterol but also other sterols found in *Drosophila* (e.g., ergosterol). For protein and sterol assays, *Spiroplasma* were removed from hemolymph samples by centrifugation at 15,000×*g* for 15 min. For whole fly analyses, 10 flies were homogenized in 250 μl PBS and the quantifications are given per fly. Note that there was no significant difference in the mass of virgin flies with and without *Spiroplasma* at 7 and 14 days of age (unpublished data). There was also no significant difference in the feeding rate of virgin flies with and without *Spiroplasma* (unpublished data) as measured by CAFE assay (*Ja et al., 2007*) over a period of 4 hr at 3, 10, and 12 days of age.

## MALDI-ToF-MS and MS/MS

*Drosophila* hemolymph (collected as described for metabolite analyses) was diluted in 1:10 in H$_2$O. Lipids were extracted using a mixture of chloroform/methanol (50/50 vol:vol). After mixing and centrifugation, 1 μl of the lipid extract was diluted with 1 μl of 9-aminoacridine matrix (25 mg/ml, dissolved in isopropanol/acetonitrile [3/2, vol/vol]). For MALDI-Tof-MS analysis, 1 μl of the mixture was rapidly spotted on an MTP 384 polished steel MALDI plate (Bruker Daltonics, Billerica, MA, USA). MALDI-Tof mass spectra of the *Drosophila* hemolymph samples were acquired using FlexControl 3.0 (Bruker Daltonics) on a Bruker AutoFlex III Smartbeam (Bruker Daltonics) in a negative reflectron mode at a laser beam attenuation of 50 and focus of 40 at 100 Hz as laser repetition rate. A total of 1000 shots were acquired in the mass range of 400 to 2000 *m/z*. Data were processed with FlexAnalysis 3.0 (Bruker Daltonics). Calibration of the instrument was performed using Peptide Standard Calibration II (Bruker Daltonics). 1000 ion counts were accumulated in the mass spectrometer prior to fragmentation. Fragmentation was performed with the Bruker designed method and at a laser beam attenuation of 42%, a laser repetition rate of 100 Hz and a reflector detector voltage set to 1.861 kV in a negative mode.

## LC-MS/MS

Hemolymph was extracted from 1250 7-day-old adult *Spiroplasma*-infected and *Spiroplasma*-uninfected mated female flies by pricking flies in the abdomen and thorax before centrifugation of flies at 16000×*g* at 4°C for 30 min in a 10-μm filter spin column (Mobitec, Goettingen, Germany). Hemolymph was subsequently filtered in Ultrafree-MC Centrifugal Filter Units (0.22 μm pores) at 16000×*g* at 4°C for 10 min. These samples were then sent to Avanti Polar Lipids analytical services division (http://www.avantilipids.com) where lipids were extracted (*Folch et al., 1957*) prior to extracts being spiked with internal standards for LC-MS/MS quantification of diacylglygerol species concentrations (*Hutchins et al., 2008*).

## Data treatment and statistical analysis

Statistical significance was calculated using a Gehan-Breslow-Wilcoxon test for survivals and an unpaired Student's $t$ test for all other experiments (with GraphPad Prism 5.0) and considered significant if p-values were lower than 0.05. Asterisks indicate the level of significance: $*p<0.05$, $**p<0.01$, and $***p<0.001$ and NS (non-significant).

## Acknowledgements

We would like to thank Laure Beven, Wilhelm Palm, Barbara Herren, Roshan Vijendravarma, Steve Perlman, and Melanie Blokesch for informative discussions and suggestions regarding the manuscript. We would like to thank Suzanne Eaton and Ronald P Kühnlein for donating fly stocks and Ryma Acheraiou for assistance. We thank the TRiP at Harvard Medical School (NIH/NIGMS R01-GM084947) and the Bloomington stock center for providing transgenic RNAi fly stocks used in this study. This work was supported by an ERC Advanced Grant.

## Additional information

### Funding

| Funder | Grant reference number | Author |
| --- | --- | --- |
| European Research Council | 339970 | Juan C Paredes, Fanny Schüpfer, Bruno Lemaitre |

The funder had no role in study design, data collection and interpretation, or the decision to submit the work for publication.

### Author contributions

JKH, Conceiving the project, Design of experiments, Performing experiments, Analyzing data, Drafting of article; JCP, Design of experiments, Performing experiments, Analysis of data, Drafting of article; FS, KA, PB, Performing experiments, Analysis of data; BL, Conceiving the project, Design of experiments, Drafting of article

### Author ORCIDs

Jeremy K Herren, 🆔 http://orcid.org/0000-0003-2239-7275

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
