## [Decision Letter]

Thank you for sending your work entitled “Regulation of Insect Endosymbiont Proliferation by Lipid Availability” for consideration at *eLife*. Your article has been evaluated by Richard Losick (Senior editor), a Reviewing editor, and 3 reviewers.

The Reviewing editor and the other reviewers discussed their comments before we reached the decision that some major revisions are needed before this manuscript is accepted for publication. The Reviewing editor has assembled the following comments to help you prepare a revised submission.

Although the experiments have been done well and the reviewers are largely satisfied with the quality of data, every reviewer is concerned with the interpretations and the underlying model presented in this paper. Each reviewer pointed out that there is no evidence presented for an adaptive mechanism and the results can easily be explained as representing a control of bacterial growth depending upon the limitations of lipid supply provided by the fly. In the revised version, please make sure that this discussion is reversed in the abstract and the main article. It may be OK to provide a clearly stated speculation at the end, but the idea that the data point to fitness needs to be modified. Please do take this suggestion seriously as the reviewers were quite unconvinced by the arguments.

We summarize some of the comments of the reviewers along these lines here:

1) The authors interpret their results in terms of a parasite strategy to limits its own titer to preserve the fitness of the flies: “... *Spiroplasma* proliferation is optimized to minimize fitness costs to the host under conditions of nutrient limitation.” None of the reviewers agree with this conclusion. In fact, the reviewers all agree that this likely happens for any internal parasite uniquely dependent on its host for growth. Also, the reviewers do not agree with the title of the manuscript as it offers no proof of regulation. One way to address the question would be to manipulate the parasite, something that is not feasible at present. One reviewer suggested that the authors could use *Spiroplasma citri*, a parasite that supposedly cannot control its own growth. The authors may wish to consider this suggestion, but it may be simpler to reword the title and manuscript.

2) Is there a control mechanism? Titers continuously increase with age of flies as shown in Figure 1 – up to the last day sampled (day 28) in flies on rich diets. If symbionts become ever denser until the fly dies at an old age, it's hard to argue that titer is adaptively regulated to optimize a tradeoff between host fitness and symbiont transmission (as hypothesized). It would appear instead that they start at a low titer and are expanding as the fly ages but never reaching their “limit” imposed by the DAG levels. Or maybe the point is that such a limit is only imposed on poor diets. On these, it does appear that the titers level off. Since intracellular bacteria can have very long doubling times, maybe these are only starting to reach their limit after 21 days.

3) The experiments show that a limit is imposed when DAG is low. But would a higher titer be reached when DAG is increased above that achieved by normal flies on rich diet? If DAG could be experimentally elevated without altering fly diet, it would be interesting to see whether the symbionts increase faster across the lifespan.

Even if that cannot be tested, it is still meaningful that on poor diets, DAG contributes to controlling numbers. The arguments about the adaptiveness of using DAG rather than glucose or trehalose make sense, but are not really proof of an adaptive control. Maybe *Spiroplasma* is limited by lipids and reaches ecological carrying capacity, and this happens to be at a titer that does not kill the fly but may still not be optimal for the “trade-off”. The paper is claiming this mechanism that has evolved to achieve some optimal level of proliferation (e.g., in the Abstract) but maybe has only shown that *Spiroplasma* is limited by lipids.

4) The experiments claiming that poor diet reduces bacterial growth and pathogenesis are over interpreted, at best. The diet is so poor that these flies live a max of 15 days, with mortality starting at day 2 or 3. So, this diet is basically lethal and it is unclear is much can be learned from this assay. The authors should consider removing this part from the paper or provide alternative conditions. The connection between DAG to cardiolipin is not developed and needs better clarification and explanations.

5) The choice of performing this study on virgins is not discussed and justified in any manner. As egg production requires the bulk of lipids, one wonders what happens to egg production in mated females under nutrient-poor conditions and how the parasite is able to optimize its transmission under these adverse conditions. What happens then to oogenesis and parasite transmission? If there are not enough lipids around to sustain oogenesis, will the parasite titer remain low so as to benefit should conditions ultimately improve? Clearly, it would be essential to monitor egg production as well as parasite transmission under these nutrient-poor conditions (which might be actually too harsh given the premature death of flies; flies given only a sucrose diet live much longer), and not just parasite titer and host longevity.

Of note, even virgin females do lay eggs after a few days. These represent also a loss of lipids. What happens under normal conditions: is there a difference in egg production between infected and non-infected flies? Are the virgin flies still laying eggs under nutrient-poor conditions?

---

## [Author Response]

*Although the experiments have been done well and the reviewers are largely satisfied with the quality of data, every reviewer is concerned with the interpretations and the underlying model presented in this paper. Each reviewer pointed out that there is no evidence presented for an adaptive mechanism and the results can easily be explained as representing a control of bacterial growth depending upon the limitations of lipid supply provided by the fly. In the revised version, please make sure that this discussion is reversed in the abstract and the main article. It may be OK to provide a clearly stated speculation at the end, but the idea that the data point to fitness needs to be modified. Please do take this suggestion seriously as the reviewers were quite unconvinced by the arguments*.

We agree with the statement our results can “be explained as representing a control of bacterial growth depending upon the limitations of lipid supply by the fly”, and do not fully understand the contradiction between this statement and our suggestion of an adaptive regulatory mechanism. However, we are in agreement that we do not provide evidence for an adaptive mechanism (this work was more focused on molecular and metabolic interactions and less on evolutionary interactions). We have now removed any discussion of a lipid-dependence based adaptation from the main text and Abstract.

In the conclusion we speculate that the mechanisms uncovered could have an adaptive function. We strived to use wording that emphasizes that this remains a speculation, e.g. “We hypothesize that *Spiroplasma’s* dependence on hemolymph lipid availability could be of adaptive significance”.

We summarize some of the comments of the reviewers along these lines here:

*1) The authors interpret their results in terms of a parasite strategy to limits its own titer to preserve the fitness of the flies: “...* Spiroplasma *proliferation is optimized to minimize fitness costs to the host under conditions of nutrient limitation.” None of the reviewers agree with this conclusion. In fact, the reviewers all agree that this likely happens for any internal parasite uniquely dependent on its host for growth. Also, the reviewers do not agree with the title of the manuscript as it offers no proof of regulation. One way to address the question would be to manipulate the parasite, something that is not feasible at present. One reviewer suggested that the authors could use* Spiroplasma citri*, a parasite that supposedly cannot control its own growth. The authors may wish to consider this suggestion, but it may be simpler to reword the title and manuscript*.

We believe our results are consistent with a “parasite strategy to preserve the fitness of the flies” but as highlighted above we have not proven this. We agree with the reviewers’ opinions regarding this matter and have made significant changes to the manuscript that reflect this. For example, we no longer use the term ‘regulation’, instead suggesting that lipid availability ‘limits’ the proliferation of *Spiroplasma*. The title of the manuscript has been changed from “Regulation of Insect Endosymbiont Proliferation by 2 Lipid Availability” to “Insect Endosymbiont Proliferation is limited by Lipid Availability”.

We are not in complete agreement with the notion that “this likely happens for any internal parasite uniquely dependent on its host for growth”. We would suggest that internal parasites can differ in their metabolic capacities and those that are able to make lipids from other diverse (and potentially more abundant) host metabolites would be expected to have a proliferation rate that is less specifically coupled to the composition of host hemolymph. In conclusion, we do agree that at some level the proliferation of an internal parasite will be limited by availability of host nutrients, but that this level differs between parasites (based on their needs/metabolic capacities) and this difference can have important implications.

*2) Is there a control mechanism? Titers continuously increase with age of flies as shown in*
Figure 1
*– up to the last day sampled (day 28) in flies on rich diets. If symbionts become ever denser until the fly dies at an old age, it's hard to argue that titer is adaptively regulated to optimize a tradeoff between host fitness and symbiont transmission (as hypothesized). It would appear instead that they start at a low titer and are expanding as the fly ages but never reaching their “limit” imposed by the DAG levels. OR maybe the point is that such a limit is only imposed on poor diets. On these, it does appear that the titers level off. Since intracellular bacteria can have very long doubling times, maybe these are only starting to reach their limit after 21 days*.

We have now provided a closer investigation into the late-timepoint interactions between *Spiroplasma* proliferation and hemolymph DAG levels. We do indeed find that *Spiroplasma* titers reach a limit in old flies (Figure 1). We also found very low levels of DAG in these older flies (Figure 5—figure supplement 1) and therefore our data is consistent with a DAG mediated limitation of *Spiroplasma* proliferation in old flies. As such, we have identified a second condition where reduced *Spiroplasma* proliferation coincides with reduced hemolymph DAG. We note that this late reduction of *Spiroplasma* proliferation does not appear to be a very effective control strategy, since there is high mortality of flies over this period. The MSRO strain of *Spiroplasma* used in this study achieves titers higher than most all other *Drosophila* endosymbiotic *Spiroplasmas* (29) and therefore we suggest that old age regulation might be “too little too late” in this case, but could play a more important role for some other *Spiroplasma* strains. As a side note we are aware that the high mortality has the potential to bias the results for the late (35 day) timepoint analysis of *Spiroplasma* titers in Figure 1. For the 35 day timepoint, we did however specifically separate flies that appeared to be very close to dying and those that were in an apparently much more healthy state and there was no significant difference between the two (data not shown).

*3) The experiments show that a limit is imposed when DAG is low. But would a higher titer be reached when DAG is increased above that achieved by normal flies on rich diet? If DAG could be experimentally elevated without altering fly diet, it would be interesting to see whether the symbionts increase faster across the lifespan*.

Experimentally (or directly) increasing DAG levels is very challenging due to the high 3 insolubility of these molecules in aqueous solutions. We were therefore unable to find a meaningful way of increasing hemolymph DAGs. Several preliminary experiments in which DAG dissolved in ethanol were injected into flies resulted in very high levels of fly mortality.

*Even if that cannot be tested, it is still meaningful that on poor diets, DAG contributes to controlling numbers. The arguments about the adaptiveness of using DAG rather than glucose or trehalose make sense, but are not really proof of an adaptive control. Maybe* Spiroplasma *is limited by lipids and reaches ecological carrying capacity, and this happens to be at a titer that does not kill the fly but may still not be optimal for the “trade-off”. The paper is claiming this mechanism that has evolved to achieve some optimal level of proliferation (e.g., in the Abstract) but maybe has only shown that* Spiroplasma *is limited by lipids*.

We accept the reviewers’ suggestion that the carrying capacity of *Spiroplasma* in the hemolymph is determined by lipid availability. We believe this is an important and interesting finding, central to this piece of work. We are in agreement that this manuscript offers no proof for adaptive control. We have done our best to identify all the parts of the manuscript that might have suggested otherwise and to modify / remove them. We think that the concept of adaptive regulation by lipid dependence is interesting and merits being mentioned as a hypothesis in the Discussion.

*4) The experiments claiming that poor diet reduces bacterial growth and pathogenesis are over interpreted, at best. The diet is so poor that these flies live a max of 15 days, with mortality starting at day 2 or 3. So, this diet is basically lethal and it is unclear is much can be learned from this assay. The authors should consider removing this part from the paper or provide alternative conditions. The connection between DAG to cardiolipin is not developed and needs better clarification and explanations*.

We have taken these comments into consideration; however, we still believe that the experiments carried out on the poor diet do offer important insights. We do not agree that the poor diet is ‘basically lethal’, since it can support the entire life cycle of *Drosophila* (a point we have now added to the Results section.) The study that developed this diet used it to maintain *Drosophila* larvae for and experimental evolution study. We reason that in nature, *Drosophila* are exposed to a range of conditions, at one extreme nutrient deprivation (our poor diet) and at the other nutrient abundance (our rich diet), probably going between the two over their lifetime. We agree that the implications of nutrient deprivation can be complex and therefore we have been cautious not to overinterpret the findings based on these experiments. We would like to stress that the findings made under poor nutritive conditions are presented together with numerous experiments under rich conditions as support for more general picture of the metabolic interactions between *Spiroplasma* and *Drosophila*.

Regarding the connection between DAG to cardiolipin, we have provided some clarifications, including a description of the molecular components of DAGs and cardiolipins in the Results section.

5) The choice of performing this study on virgins is not discussed and justified in any manner. As egg production requires the bulk of lipids, one wonders what happens to egg production in mated females under nutrient-poor conditions and how the parasite is able to optimize its transmission under these adverse conditions. What happens then to oogenesis and parasite transmission? If there are not enough lipids around to sustain oogenesis, will the parasite titer remain low so as to benefit should conditions ultimately improve? Clearly, it would be essential to monitor egg production as well as parasite transmission under these nutrient-poor conditions (which might be actually too harsh given the premature death of flies; flies given only a sucrose diet live much longer), and not just parasite titer and host longevity. Of note, even virgin females do lay eggs after a few days. These represent also a loss of lipids. What happens under normal conditions: is there a difference in egg production between infected and non-infected flies? Are the virgin flies still laying eggs under nutrient-poor conditions?

The principal reason for carrying out this study on virgins is that because the *Spiroplasma* strain used causes male killing (and therefore all-female broods) it was more feasible to maintain *Spiroplasma*-infected and uninfected virgin flies under identical conditions than non-virgin flies. Firstly, to generate ‘identical’ non-virgin flies one has to control the timing/frequency of mating (since, when *Spiroplasma*-infected flies emerge they stay virgin, in contrast to *Spiroplasma*-uninfected flies which have the opportunity to mate). This requires collecting virgin flies (*Spiroplasma*-infected and uninfected) and then mating them with the same number of males for the same period.

Secondly, also due to the effects of *Spiroplasma*-induced male killing, there are (about 2x) more larvae in the media of *Spiroplasma*-uninfected mated flies than *Spiroplasma* infected mated flies, larval feeding activity is likely to change the composition of the media, and therefore for comparisons between mated *Spiroplasma*-infected and uninfected we therefore had to put flies on new media every 2 days. We have now carried out some of the key experiments of the paper in non-virgins (survivals, DAG hemolymph quantification) and found that these also support the conclusions drawn based on experiments carried out with virgins. In the future it would be interesting to carry out such experiments with non-male killing *Spiroplasmas* that are native to *D. melanogaster* (and also in males). So far, although there are reports of non-male killing *Spiroplasmas* native to *Drosophila melanogaster* (Watts et al., 2009, PLOS ONE), they have not been successfully maintained in the lab.

We have carried out a number of additional experiments to address other issues highlighted by the reviewers above:

We have investigated egg production by mated and unmated females on rich and poor media. The important findings of this investigation are: i) that *Spiroplasma* increases egg laying by virgins on rich media (Figure 1) and mated flies on nutrient poor media (Figure 2—figure supplement 1). ii) *Spiroplasma* increases the early production of eggs by mated flies on rich media Figure 1—figure supplement 1).

These *Spiroplasma*-mediated affects on reproduction are likely to have important consequences for fly physiology. Indeed trade-offs are known to exist between reproductive output and lifespan and this aspect of *Spiroplasma*-host biology should be investigated in future work. We note that these effects on egg production are probably not underlying many of the principal effects of *Spiroplasma* infection characterized in this paper. Notably mated flies (*Spiroplasma*-infected and uninfected) produced equivalent numbers of eggs over 2 weeks and yet we still observe a decrease in lifespan and hemolymph DAG in these flies in the presence of *Spiroplasma* (as was previously shown with virgin flies). The experiments on egg production also support our original conclusions (based on survivals) that *Spiroplasma* has rather minor detrimental effects (and indeed under some conditions possibly positive effects) on host fitness. Endosymbiont titers have been linked to transmission efficiency, and therefore (as suggested by the reviewer) one could speculate that the decrease in *Spiroplasma* titers under nutrient deprivation could have effects on *Spiroplasma* transmission. While interesting, investigating this under our experimental conditions was not feasible because so few eggs were laid per fly under conditions of nutrient deprivation. It is also notable that in mated flies, where more eggs were laid under nutrient poor conditions, the production of eggs stopped after 4-6 days and the decline in *Spiroplasma* titers under nutrient deprivation are only observed after 8 days. If transmission were to decline after 8 days on nutrient poor media, then the early increase in egg production induced by *Spiroplasma* under these conditions would make evolutionary sense. We also quantified *Spiroplasma* titers in virgins and mated flies after 14 days on rich media and found no significant difference (unpublished data). This would suggest that Spiroplasma titers are not directly linked to the rate of egg production (since virgins lay many fewer eggs that mated flies).